# Chemical Constituents from the Leaves of *Ligustrum robustum* and Their Bioactivities

**DOI:** 10.3390/molecules28010362

**Published:** 2023-01-02

**Authors:** Shi-Hui Lu, Hao-Jiang Zuo, Jing Huang, Wei-Neng Li, Jie-Lian Huang, Xiu-Xia Li

**Affiliations:** 1College of Pharmacy, Youjiang Medical University for Nationalities, Baise 533000, China; 2Department of Laboratory Science of Public Health, West China School of Public Health, Sichuan University, Chengdu 610041, China; 3Key Laboratory of Drug Targeting, Ministry of Education, West China School of Pharmacy, Sichuan University, Chengdu 610041, China; 4Nursing School, Youjiang Medical University for Nationalities, Baise 533000, China

**Keywords:** *Ligustrum robustum*, hexenol glycoside, butenol glycoside, sugar ester, FAS, α-glucosidase, antioxidant, antiobesity, hypoglycemic

## Abstract

The leaves of *Ligustrum robustum* have been consumed as Ku-Ding-Cha for clearing heat and removing toxins, and they have been used as a folk medicine for curing hypertension, diabetes, and obesity in China. The phytochemical research on the leaves of *L. robustum* led to the isolation and identification of two new hexenol glycosides, two new butenol glycosides, and five new sugar esters, named ligurobustosides X (**1a**), X_1_ (**1b**), Y (**2a**), and Y_1_ (**2b**) and ligurobustates A (**3a**), B (**3b**), C (**4b**), D (**5a**), and E (**5b**), along with seven known compounds (**4a** and **6**–**10**). Compounds **1**–**10** were tested for their inhibitory effects on fatty acid synthase (FAS), α-glucosidase, and α-amylase, as well as their antioxidant activities. Compound **2** showed strong FAS inhibitory activity (IC_50_ 4.10 ± 0.12 μM) close to that of the positive control orlistat (IC_50_ 4.46 ± 0.13 μM); compounds **7** and **9** revealed moderate α-glucosidase inhibitory activities; compounds **1**–**10** showed moderate α-amylase inhibitory activities; and compounds **1** and **10** displayed stronger 2,2′-azino-bis(3-ethylbenzthiazoline-6-sulphonic acid) ammonium salt (ABTS) radical scavenging effects (IC_50_ 3.41 ± 0.08~5.65 ± 0.19 μM) than the positive control l-(+)-ascorbic acid (IC_50_ 10.06 ± 0.19 μM). This study provides a theoretical foundation for the leaves of *L. robustum* as a functional tea to prevent diabetes and its complications.

## 1. Introduction

Diabetes, which affects nearly 10.5% of the population worldwide, is a chronic metabolic disease characterized by hyperglycemia caused by insulin resistance, a deficiency in insulin secretion, or both [1]. Its complications, including diabetic neuropathy, nephropathy, and cardiovascular diseases, lead to serious morbidity and mortality [1]. Current drugs, such as insulin, metformin, sulfonylureas, and acarbose, can control hyperglycemia, but their effect on preventing the complications of diabetes is not ideal. Therefore, it is significant to search for new resources for the prevention of diabetes and its complications. 

Studies have revealed that long-term obesity might trigger specific metabolic disorders, such as cardiovascular diseases, insulin resistance, and diabetes [2,3]; fatty acid synthase (FAS), which catalyzes the synthesis of saturated long-chain fatty acids, is a potential target to prevent obesity [4]; carbohydrate digestive enzymes, such as α-glucosidase and α-amylase, play a crucial role in promoting hyperglycemia by releasing monosaccharides in the course of digestion [5]; and the contribution of reactive oxygen species generated by oxidative stress induced by chronic hyperglycemia has been linked to the onset and progression of diabetes and its complications [6]. Thus, natural products with inhibitory activities on FAS, α-glucosidase, and α-amylase as well as an antioxidant effect might be a new resource to prevent diabetes and its complications.

*Ligustrum robustum* (Roxb.) Blume is a plant of Oleaceae, and it is distributed extensively in Southwest China, India, Burma, Vietnam, and Cambodia [4]. The leaves of *L. robustum* have been used for Ku-Ding-Cha, a tea with functions in clearing heat and removing toxins, in China since the Dong Han Dynasty [7,8]. In addition, *L. robustum* is believed as a folk medicine for curing hypertension, diabetes, obesity, etc. [8,9]. In the previous studies on *L. robustum* [4,7,8,9,10,11,12,13,14,15,16,17,18,19], more than 70 chemical ingredients, including monoterpenoid glycosides, iridoid glycosides, phenylethanoid glycosides, phenylmethanoid glycosides, flavonoid glycosides, lignan glycosides, and triterpenoids were reported. The antiobesity, anti-inflammatory, and antioxidative activities of the extract; the inhibitory effects on α-glucosidase, α-amylase, and FAS; and the antioxidant effects of some compositions were also discovered. In order to further determine the active constituents for preventing diabetes and its complications, phytochemical and biological research on the leaves of *L. robustum*, which was carried out preliminarily [4,15,16], was further performed. As a result, two new hexenol glycosides, two new butenol glycosides, and five new sugar esters, named ligurobustosides X (**1a**), X_1_ (**1b**), Y (**2a**), and Y_1_ (**2b**) and ligurobustates A (**3a**), B (**3b**), C (**4b**), D (**5a**), and E (**5b**), along with seven reported compounds (**4a** and **6**–**10**) (Figure 1), were isolated and identified from the leaves of *L. robustum*. This paper reports the isolation and structural identification of compounds **1**–**10** and describes their inhibitory activities on FAS, α-glucosidase, and α-amylase and their antioxidant effects.

## 2. Results and Discussion

### 2.1. Identification of Compounds ***1***–***10***

Compound **1** was obtained as a white amorphous powder, and its molecular formula was analyzed as C_27_H_38_O_12_ by HRESIMS (*m/z* 577.2260 [M + Na]^+^, calculated 577.2261 for C_27_H_38_NaO_12_). The NMR spectra of **1** showed two stereoisomers: **1a** and **1b** (5:3). In the ^1^H NMR spectrum of **1a** (Table 1), the following signals were observed: (1) a 4-substituted phenyl at *δ*_H_ 6.77, 7.43 (2H each, d, *J* = 8.4 Hz); (2) two trans double bonds at *δ*_H_ 6.33, 7.63 (1H each, d, *J* = 15.6 Hz) and 5.36, 5.42 (1H each, dt, *J* = 17.4, 6.6 Hz); (3) two anomeric protons at *δ*_H_ 4.31 (1H, d, *J* = 8.4 Hz) and 5.18 (1H, d, *J* = 1.8 Hz); (4) a methylene linking with oxygen at *δ*_H_ 3.55, 3.80 (1H each, m), two methylene groups at *δ*_H_ 2.05, 2.37 (2H each, m), and two methyl groups at *δ*_H_ 0.93 (2H, t, *J* = 7.2 Hz, 6a), 0.97 (1H, t, *J* = 7.2 Hz, 6b) and 1.25 (3H, d, *J* = 6.0 Hz). In the ^13^C NMR spectrum of **1a** (Table 2), the following signals were observed: a carbonyl at *δ*_C_ 169.2, a phenyl at *δ*_C_ 117.4–163.0, two double bonds at *δ*_C_ 114.1–147.1, two anomeric carbons at *δ*_C_ 102.7 and 104.4, nine sugar carbons at *δ*_C_ 64.6–84.0, a methylene linking with oxygen at *δ*_C_ 70.8, two methylene groups at *δ*_C_ 21.5 and 28.9, and two methyl groups at *δ*_C_ 14.6 and 17.9. The above ^1^H and ^13^C NMR data suggested **1a** should be a glycoside, including a trans-*p*-coumaroyl and two monosaccharide moieties. The ^1^H-^1^H COSY experiment of **1a** (Figure 2) showed correlations between *δ*_H_ 2.37 (H-2 of aglycone) and *δ*_H_ 3.80 (H-1b of aglycone); 5.36 (H-3 of aglycone) between *δ*_H_ 5.36 (H-3 of aglycone) and *δ*_H_ 5.42 (H-4 of aglycone); between *δ*_H_ 2.05 (H-5 of aglycone) and *δ*_H_ 5.42 (H-4 of aglycone), 0.93 (H-6a of aglycone). Together with the HMBC experiment on **1a** (Figure 2), the aglycone of **1a** was affirmed as (*E*)-3-hexen-1-ol. The acid hydrolysis experiment of **1** resulted in d-glucose and l-rhamnose, affirmed by TLC and a comparison of its NMR data with those of ligurobustoside E [12]. The HMBC experiment on **1a** (Figure 2) displayed the following long-distance correlations: between *δ*_H_ 4.31 (H-1′ of glucosyl) and *δ*_C_ 70.8 (C-1 of aglycone), between *δ*_H_ 5.18 (H-1″ of rhamnosyl) and *δ*_C_ 84.0 (C-3′ of glucosyl), and between *δ*_H_ 4.35 (H-6′a of glucosyl), 4.48 (H-6′b of glucosyl), and *δ*_C_ 169.2 (carbonyl of coumaroyl). The ^1^H and ^13^C NMR signals of **1** were assigned by ^1^H-^1^H COSY, HSQC, and HMBC experiments (Appendix A). Based on above evidence, **1a** was identified as (*E*)-3-hexen-1-yl 3-*O*-(α-l-rhamnopyranosyl)-6-*O*-(*trans*-*p*-coumaroyl)-*O*-β-d-glucopyranoside. It is a novel hexenol glycoside, named ligurobustoside X. 

The NMR data of **1b** (Table 1 and Table 2) were similar to those of **1a**, except the *trans*-*p*-coumaroyl in **1a** was replaced by the *cis*-*p*-coumaroyl (*δ*_H_ 5.79, 6.88 (1H each, d, *J* = 13.2 Hz, H-8′″, H-7′″)) in **1b**. The HMBC experiment on **1b** (Figure 2) displayed long-distance correlations between *δ*_H_ 4.27 (H-1′ of glucosyl) and *δ*_C_ 70.7 (C-1 of aglycone), between *δ*_H_ 5.16 (H-1″ of rhamnosyl) and *δ*_C_ 84.0 (C-3′ of glucosyl), and between *δ*_H_ 4.34 (H-6′a of glucosyl), 4.46 (H-6′b of glucosyl), and *δ*_C_ 168.1 (carbonyl of coumaroyl). Therefore, the structure of compound **1b** was identified as (*E*)-3-hexen-1-yl 3-*O*-(α-l-rhamnopyranosyl)-6-*O*-(*cis*-*p*-coumaroyl)-*O*-β-d-glucopyranoside. It is a novel hexenol glycoside, named ligurobustoside X_1_. In conclusion, compound **1** is a mixture of ligurobustosides X and X_1_.

Compound **2** was obtained as a white amorphous powder, and its molecular formula was determined as C_25_H_34_O_12_ by HRESIMS (*m/z* 549.1941 [M + Na]^+^, calculated 549.1948 for C_25_H_34_NaO_12_). The NMR spectra of **2** showed two stereoisomers: **2a** and **2b** (2:1). In the ^1^H NMR spectrum of **2a** (Table 1), the following signals were revealed: (1) a 4-substituted phenyl at *δ*_H_ 6.80 and 7.47 (2H each, d, *J* = 8.4 Hz); (2) a trans double bond at *δ*_H_ 6.37 and 7.65 (1H each, d, *J* = 16.2 Hz); (3) two olefinic proton signals at *δ*_H_ 4.88 and 5.02 (1H each, br. s); (4) two anomeric protons at *δ*_H_ 4.30 (1H, d, *J* = 7.2 Hz) and 5.18 (1H, d, *J* = 1.8 Hz); (5) a methylene linking with oxygen at *δ*_H_ 4.07 and 4.20 (1H each, d, *J* = 12.6 Hz); and two methyl groups at *δ*_H_ 1.75 (3H, s) and 1.25 (3H, d, *J* = 6.6 Hz). In the ^13^C NMR spectrum of **2a** (Table 2), the following signals were shown: a carbonyl at *δ*_C_ 169.1, a phenyl at *δ*_C_ 116.9–161.6, two double bonds at *δ*_C_ 113.4–146.9, two anomeric carbons at *δ*_C_ 102.8 and 103.0, nine sugar carbons at *δ*_C_ 64.6–84.0, a methylene linking with oxygen at *δ*_C_ 74.0, and two methyl groups at *δ*_C_ 17.9 and 19.7. The above ^1^H and ^13^C NMR data indicated that **2a** should be a glycoside, including a *trans*-*p*-coumaroyl and two monosaccharide moieties. In the HMBC experiment on **2a** (Figure 2), the following long-distance correlations were displayed: between *δ*_H_ 4.07 (H-1a of aglycone) and 4.20 (H-1b of aglycone) and *δ*_C_ 143.1 (C-2 of aglycone), 113.4 (C-3 of aglycone), and 19.7 (C-4 of aglycone); between *δ*_H_ 4.88 (H-3a of aglycone), 5.02 (H-3b of aglycone), and *δ*_C_ 19.7 (C-4 of aglycone). Together with the HSQC experiment on **2a** (Appendix A), the aglycone of **2a** was affirmed as 2-methyl-2-propen-1-ol. The acid hydrolysis experiment on **2** afforded d-glucose and l-rhamnose, confirmed by TLC and a comparison of its NMR data with those of ligurobustoside E [12]. Furthermore, the HMBC experiment on **2a** (Figure 2) displayed the following long-distance correlations: between *δ*_H_ 4.30 (H-1′ of glucosyl) and *δ*_C_ 74.0 (C-1 of aglycone), between *δ*_H_ 5.18 (H-1″ of rhamnosyl) and *δ*_C_ 84.0 (C-3′ of glucosyl), and between *δ*_H_ 4.36 (H-6′a of glucosyl), 4.48 (H-6′b of glucosyl), and *δ*_C_ 169.1 (carbonyl of coumaroyl). The ^1^H and ^13^C NMR signals of **2** were assigned by ^1^H-^1^H COSY, HSQC, and HMBC experiments (Appendix A). Thus, the structure of **2a** was elucidated as 2-methyl-2-propen-1-yl 3-*O*-(α-l-rhamnopyranosyl)-6-*O*-(*trans*-*p*-coumaroyl)-*O*-β-d-glucopyranoside. It is a novel butenol glycoside, named ligurobustoside Y.

The NMR data of **2b** (Table 1 and Table 2) were similar to those of **2a**, except the *trans*-*p*-coumaroyl in **2a** was replaced by the *cis*-*p*-coumaroyl (*δ*_H_ 5.80, 6.89 (1H each, d, *J* = 12.6 Hz, H-8′″, H-7′″)) in **2b**. In the HMBC experiment on **2b** (Figure 2), the following long-distance correlations were observed: between *δ*_H_ 4.26 (H-1′ of glucosyl) and *δ*_C_ 73.8 (C-1 of aglycone), between *δ*_H_ 5.16 (H-1″ of rhamnosyl) and *δ*_C_ 84.0 (C-3′ of glucosyl), and between *δ*_H_ 4.36 (H-6′a of glucosyl), 4.46 (H-6′b of glucosyl), and *δ*_C_ 168.1 (carbonyl of coumaroyl). Therefore, the structure of **2b** was identified as 2-methyl-2-propen-1-yl 3-*O*-(α-l-rhamnopyranosyl)-6-*O*-(*cis*-*p*-coumaroyl)-*O*-β-d-glucopyranoside. It is a novel butenol glycoside, named ligurobustoside Y_1_. In summary, compound **2** is a mixture of ligurobustosides Y and Y_1_.

Compound **3** was obtained as a white amorphous powder, and its molecular formula was determined as C_21_H_28_O_12_ by HRESIMS (*m/z* 495.1474 [M + Na]^+^, calculated 495.1478 for C_25_H_34_NaO_12_). The NMR spectra of **3** exhibited two stereoisomers: **3a** and **3b** (4:1). The ^1^H and ^13^C NMR spectra of **3a** (Table 3 and Table 4) showed a *trans*-*p*-coumaroyl (*δ*_H_ 7.63, 6.33 (1H each, d, *J* = 16.2 Hz, H-7″, H-8″), 7.45 and 6.80 (2H each, d, *J* = 8.4 Hz, H-2″, H-3″, H-5″, H-6″); *δ*_C_ 126.9 (C-1″), 161.6 (C-4″), 169.2 (CO)], an α-rhamnosyl (*δ*_H_ 5.18 (1H, d, *J* = 1.8 Hz, H-1′), 1.26 (3H, d, *J* = 6.0 Hz, H-6′); *δ*_C_ 102.7 (C-1′), 17.9 (C-6′)), and a substituted glucose, which kept balance between the β and α configurations in CD_3_OD (β-configuration: *δ*_H_ 4.52 (1H, d, *J* = 7.8 Hz, H-1), *δ*_C_ 98.1 (C-1); α-configuration: *δ*_H_ 5.08 (1H, d, *J* = 3.6 Hz, H-1), *δ*_C_ 94.0 (C-1)). The acid hydrolysis experiment on **3** offered d-glucose and l-rhamnose confirmed by TLC and a comparison of its NMR data with those of ligurobustoside E [12]. The HMBC experiment on **3a** (β, Figure 2) displayed the following long-distance correlations: between *δ*_H_ 5.18 (H-1′ of rhamnosyl) and *δ*_C_ 84.1 (C-3 of glucose) and between *δ*_H_ 4.36 (H-6a of glucose), 4.45 (H-6b of glucose) and *δ*_C_ 169.2 (carbonyl of coumaroyl). The ^1^H and ^13^C NMR signals of **3** were assigned by ^1^H-^1^H COSY, HSQC and HMBC experiment (Appendix A). Based on the above evidence, the structure of compound **3a** was identified to be 3-*O*-(α-l-rhamnopyranosyl)-6-*O*-(*trans*-*p*-coumaroyl)-d-glucopyranose. It is a new sugar ester, named ligurobustate A. 

The NMR data of **3b** (Table 3 and Table 4) were close to those of **3a**. The main difference was that the *trans*-*p*-coumaroyl in **3a** was replaced by the *cis*-*p*-coumaroyl (*δ*_H_ 6.86, 5.76 (1H each, d, *J* = 13.2 Hz, H-7″, H-8″)) in **3b**. The HMBC experiment on **3b** (β, Figure 2) displayed the following long-distance correlations: between *δ*_H_ 5.15 (H-1′ of rhamnosyl) and *δ*_C_ 84.2 (C-3 of glucose) and between *δ*_H_ 4.26 (H-6a of glucose), 4.39 (H-6b of glucose), and *δ*_C_ 168.2 (carbonyl of coumaroyl). Therefore, the structure of compound **3b** was identified to be 3-*O*-(α-l-rhamnopyranosyl)-6-*O*-(*cis*-*p*-coumaroyl)-d-glucopyranose. It is a new sugar ester, named ligurobustate B. In summary, compound **3** is a mixture of ligurobustates A and B.

Compound **4**, a white amorphous powder, was determined as C_21_H_28_O_12_ by HRESIMS (*m/z* 495.1476 [M + Na]^+^, calculated 495.1478 for C_21_H_28_NaO_12_). The NMR spectra of **4** exhibited two stereoisomers: **4a** and **4b** (3:1). The ^1^H and ^13^C NMR data of **4a** (Appendix A) was in accordance with those of 3-*O*-(α-l-rhamnopyranosyl)-4-*O*-(*trans*-*p*-coumaroyl)-d-glucopyranose (cistanoside I) [20]. The NMR data of **4b** (Table 3 and Table 4) were similar to those of **4a**, except the *trans*-*p*-coumaroyl (*δ*_H_ 7.67, 6.35 (1H each, d, *J* = 16.0 Hz, H-7″, H-8″)) in **4a** was replaced by the *cis*-*p*-coumaroyl (*δ*_H_ 6.94, 5.81 (1H each, d, *J* = 12.8 Hz, H-7″, H-8″)) in **4b**. The acid hydrolysis experiment on **4** resulted in d-glucose and l-rhamnose, confirmed by TLC. The HMBC experiment on **4b** (β, Figure 2) showed the following long-distance correlations: between *δ*_H_ 5.12 (H-1′ of rhamnosyl) and *δ*_C_ 81.9 (C-3 of glucose), and between *δ*_H_ 4.85 (H-4 of glucose) and *δ*_C_ 167.0 (carbonyl of coumaroyl). The ^1^H and ^13^C NMR signals of **4** were assigned by ^1^H-^1^H COSY, HSQC, and HMBC experiments (Appendix A). Thus, **4b** was identified as 3-*O*-(α-l-rhamnopyranosyl)-4-*O*-(*cis*-*p*-coumaroyl)-d-glucopyranose. It is a new sugar ester, named ligurobustate C. To sum up, compound **4** is a mixture of cistanoside I and ligurobustate C.

Compound **5**, a white amorphous powder, was analyzed as C_27_H_38_O_16_ by HRESIMS (*m/z* 641.2057 [M + Na]^+^, calculated 641.2058 for C_27_H_38_NaO_16_). The NMR spectra of **5** showed two stereoisomers: **5a** and **5b** (5:1). The NMR data of **5a** (Table 3 and Table 4) were close to those of **3a**, except for another α-rhamnosyl (*δ*_H_ 5.19 (1H, d, *J* = 1.6 Hz, H-1′), 1.29 (3H, d, *J* = 6.0 Hz, H-6′); *δ*_C_ 102.4 (C-1′), 18.6 (C-6′)). The acid hydrolysis experiment on **5** afforded d-glucose and l-rhamnose, affirmed by TLC and a comparison of its NMR data with those of **3**. The HMBC experiment on **5a** (β, Figure 2) revealed the following long-distance correlations: between *δ*_H_ 5.19 (H-1′ of inner rhamnosyl) and *δ*_C_ 83.6 (C-3 of glucose), between *δ*_H_ 5.20 (H-1″ of outer rhamnosyl) and *δ*_C_ 81.2 (C-4′ of inner rhamnosyl), and between *δ*_H_ 4.33 (H-6a of glucose), 4.45 (H-6b of glucose), and *δ*_C_ 169.2 (carbonyl of coumaroyl). The ^1^H and ^13^C NMR signals of **5** were assigned by ^1^H-^1^H COSY, HSQC, and HMBC experiment s(Appendix A). Based on the above evidence, **5a** was identified to be 3-*O*-[α-l-rhamnopyranosyl-(1→4)-α-l-rhamnopyranosyl]-6-*O*-(*trans*-*p*-coumaroyl)-d-glucopyranose. It is a new sugar ester, named ligurobustate D. 

The NMR data of **5b** (Table 3 and Table 4) were close to those of **5a**; the main difference was that the *trans*-*p*-coumaroyl (*δ*_H_ 7.64, 6.35 (1H each, d, *J* = 16.0 Hz, H-7″′, H-8″′)) in **5a** was replaced by the *cis*-*p*-coumaroyl (*δ*_H_ 6.87, 5.79 (1H each, d, *J* = 12.8 Hz, H-7′″, H-8′″)) in **5b**. The HMBC experiment on **5b** (β, Figure 2) showed the following long-distance correlations: between *δ*_H_ 5.17 (H-1′ of inner rhamnosyl) and *δ*_C_ 83.6 (C-3 of glucose), between *δ*_H_ 5.20 (H-1″ of outer rhamnosyl) and *δ*_C_ 81.2 (C-4′ of inner rhamnosyl), and between *δ*_H_ 4.33 (H-6a of glucose), 4.45 (H-6b of glucose), and *δ*_C_ 168.2 (carbonyl of coumaroyl). Thus, the structure of **5b** was elucidated to be 3-*O*-[α-l-rhamnopyranosyl-(1→4)-α-l-rhamnopyranosyl]-6-*O*-(*cis*-*p*-coumaroyl)-d-glucopyranose. It is a new sugar ester, named ligurobustate E. In conclusion, compound **5** is a mixture of ligurobustates D and E.

Compounds **6**–**10** (^1^H, ^13^C NMR data see Appendix A) were identified as reported 3-*O*-(α-l-rhamnopyranosyl)-4-*O*-(*trans*-caffeoyl)-d-glucopyranose (cistanoside F, **6**) [21]; kaempferol 3, 7-diglucoside (peonoside, **7**) [22]; (+)-cycloolivil 6-*O*-β-d-glucopyranoside (**8**) [23]; (*E*)-methyl *p*-hydroxycinnamate (**9a**) [24]; (*Z*)-methyl *p*-hydroxycinnamate (**9b**) [25]; and 4-hydroxyphenylethanol (**10**) [26]; by comparison with published NMR data and 2D-NMR experiments (^1^H-^1^H COSY, HSQC, and HMBC). Compounds **4a**, **6**, **7**, **8**, **9a**, **9b**, and **10** were isolated from this plant for the first time.

### 2.2. The Bioactivities of Compounds ***1***–***10***

Compounds **1**–**10** isolated from *L. robustum* were tested for their inhibitory activities on FAS, α-glucosidase, and α-amylase as well as their antioxidant effects. The results of the bioactivity assays are listed in Table 5.

(1) The FAS inhibitory activity of compound **2** (IC_50_ 4.10 ± 0.12 μM) was as strong as the positive control orlistat (IC_50_ 4.46 ± 0.13 μM), while the FAS inhibitory activities of compounds **3**–**5** and **7**–**9** (IC_50_ 6.25 ± 0.20~15.41 ± 0.42 μM) were weaker than orlistat. (2) The α-glucosidase inhibitory activities of compounds **7** and **9** were moderate and weaker than acarbose, which was used as a positive control. (3) The α-amylase inhibitory activities of compounds **1**–**10** were moderate and weaker than the positive control acarbose. (4) The 2,2-diphenyl-1-picrylhydrazyl (DPPH) radical scavenging effect of compound **6** (IC_50_ 46.66 ± 1.58 μM) were weaker than l-(+)-ascorbic acid (IC_50_ 13.66 ± 0.13 μM), which was applied as a positive control. (5) The 2,2′-azino-bis(3-ethylbenzthiazoline-6-sulphonic acid) ammonium salt (ABTS) radical scavenging effects of compounds **1** and **10** (IC_50_ 3.41 ± 0.08~5.65 ± 0.19 μM) were more potent than the positive control l-(+)-ascorbic acid (IC_50_ 10.06 ± 0.19 μM), while the ABTS radical scavenging effects of compounds **3**, **4**, **7**, and **9** (IC_50_ 8.78 ± 0.09~12.04 ± 0.08 μM) were as strong as l-(+)-ascorbic acid.

From the results of the DPPH and ABTS assays, the phenolic hydroxy group in a compound is believed to be a key factor for the antioxidant effect. Because FAS, obesity, and reactive oxygen species play vital roles in the initiation and progression of diabetes and its complications, and α-glucosidase and α-amylase are two important targets for treating diabetes [2,3,4,5,6], antioxidants **1**–**10**, which have some FAS, α-glucosidase, and α-amylase inhibitory activities, might be a part of the active constituents of *L. robustum* that prevent diabetes and its complications.

## 3. Materials and Methods

### 3.1. General Experimental Procedure

The NMR spectra were collected on a Bruker Ascend^TM^ 400 NMR spectrometer (Bruker, Germany) (^1^H at 400 MHz, ^13^C at 100 MHz) or an Agilent 600/54 Premium Compact NMR spectrometer (Agilent, Santa Clara, CA, USA) (^1^H at 600 MHz, ^13^C at 150 MHz) with CD_3_OD (**6**, **7**: CD_3_OD + DMSO-d_6_) as the solvent at 25 °C. The chemical shifts are expressed in *δ* (ppm) and tetramethylsilane (TMS) was used as an internal standard, while coupling the constants (*J*) are expressed in Hz. The UV spectrum was carried out using a UV2700 spectrophotometer (Shimadzu, Kyoto, Japan). The IR absorption spectrum was recorded with a PerkinElmer Spectrum Two FT-IR spectrometer (PerkinElmer, Waltham, MA, USA). High-resolution electrospray ionization mass spectroscopy (HRESIMS) was determined on a Waters Q-TOF Premier mass spectrometer (Waters, Milford, MA, USA). The optical rotation value was tested with an AUTOPOL VI automatic polarimeter (Rudolph, Hackettstown, NJ, USA).

Column chromatography (CC) was executed on silica gel (SiO_2_: 200–300 mesh, Qingdao Ocean Chemical Industry Co., Shandong, China), polyamide (60–90 mesh, Jiangsu Changfeng Chemical Industry Co., China), and MCI-gel CHP-20P (75–150 μm, Mitsubishi Chemical Co., Tokyo, Japan). The preparative HPLC was executed using a GL3000-300 mL system instrument (Chengdu Gelai Precision Instruments Co., Ltd., Sichuan, China) with a UV-3292 detector (running at 215 nm) and a C-18 column (particle size: 5 μm, 50 × 450 mm), eluting with MeOH-H_2_O at 30 mL/min. The TLC was carried out on precoated HPTLC Fertigplatten Kieselgel 60 F_254_ plates (Merck), which were sprayed with 10% sulfuric acid ethanolic solution or α-naphthol-sulfuric acid solution and then baked at 105 °C for 2–5 min. The UV-vis absorbance was measured with a Spark 10M microplate reader (Tecan Trading Co. Ltd., Shanghai, China) or a UV2700 spectrophotometer (Shimadzu, Kyoto, Japan). NADPH and acetyl-coenzyme A (Ac-CoA) were afforded by Zeye Biochemical Co., Ltd. (Shanghai, China). The Methylmalonyl coenzyme A tetralithium salt hydrate (Mal-CoA) was purchased from Sigma-Aldrich (St. Louis, MO, USA). 2,2′-Azino-bis(3-ethylbenzthiazoline-6-sulphonic acid) ammonium salt (ABTS) was acquired from Aladdin Industrial Co., Ltd. (Shanghai, China). 2,2-Diphenyl-1-picrylhydrazyl (DPPH) was obtained from Macklin Biochemical Co., Ltd. (Shanghai, China). 

### 3.2. Plant Material

The fresh leaves of *L. robustum* were gathered from Yibin City, Sichuan Province, China, in April 2017, and confirmed by Guo-Min Liu (Kudingcha Research Institute, Hainan University, Haikou, China). A voucher sample (No. 201704lsh) was saved at the West China School of Pharmacy, Sichuan University, Chengdu, China.

### 3.3. Extraction and Isolation

The fresh leaves of *L. robustum* were turned and heated at 120 °C for 50 min and then crushed. The crushed leaves (7.0 kg) were extracted with 70% ethanol (28 L × 1) under reflux in a multifunction extractor for 2 h [4]. The ethanol extract was filtered and condensed in vacuo to acquire a paste (2.2 kg). The paste was dissolved with 3 L 95% ethanol, and then 3 L of purified water was added to deposit the chlorophyll. After percolation, the filtrate was concentrated in vacuo to obtain a residue (1.0 kg). The residue was separated on a silica gel column (CH_2_Cl_2_-MeOH, 10:0–0:10) to offer Fr. I (84 g), Fr. II (145 g), Fr. III (93 g), and Fr. IV (70 g). Fr. II was separated twice on silica gel column (CH_2_Cl_2_-MeOH-H_2_O, 200:10:1–80:20:2; or EtOAc-MeOH-H_2_O, 100:4:2–100:6:2), isolated by CC with polyamide (EtOH-H_2_O, 0:10–6:4) and MCI (MeOH-H_2_O, 0:10–7:3), and then purified by preparative HPLC (MeOH-H_2_O, 24:76–62:38) to obtain **1** (21.5 mg), **2** (5.1 mg), **8** (53.2 mg), **9** (8.3 mg), and **10** (27.9 mg). Fr. III was separated repeatedly by CC with silica gel (EtOAc-MeOH-H_2_O, 100:4:2–100:20:10), subjected to a polyamide column (EtOH-H_2_O, 0:10–6:4) and MCI column (MeOH-H_2_O, 2:8–6:4), and then purified by preparative HPLC (MeOH-H_2_O, 20:80–40:60) and a silica gel column (EtOAc-MeOH-H_2_O, 100:4:2–100:6:3) or recrystallized in methanol to yield **3** (87.8 mg), **4** (32.8 mg), **5** (15.8 mg), **6** (32.6 mg), and **7** (6.1 mg). 

Compound **1**: white amorphous powder. [*α*]^30^_D_ −34.8 (c 0.33, MeOH); UV (MeOH) λ_max_: (log ε) 213 (4.1), 227 (4.2), 316 (4.4) nm; IR (film) ν_max_: 3380, 2927, 1692, 1604, 1514, 1446, 1269, 1168, 1089, 1038, 834 cm^–1^; ^1^H NMR (CD_3_OD, 600 MHz) data, see Table 1; ^13^C NMR (CD_3_OD, 150 MHz) data, see Table 2; HRESIMS *m/z* 577.2260 [M + Na]^+^ (calculated for C_27_H_38_NaO_12_, 577.2261).

Compound **2**: white amorphous powder. [*α*]^30^_D_ −11.8 (*c* 0.10, MeOH); UV (MeOH) λ_max_ (log ε): 213 (4.1), 226 (4.2), 317 (4.4) nm; IR (film) ν_max_: 3360, 2924, 2853, 1692, 1635, 1605, 1515, 1456, 1170, 1040 cm^–1^; ^1^H NMR (CD_3_OD, 600 MHz) data, see Table 1; ^13^C NMR (CD_3_OD, 150 MHz) data, see Table 2; HRESIMS *m/z* 549.1941 [M + Na]^+^ (calculated for C_25_H_34_NaO_12_, 549.1948).

Compound **3**: white amorphous powder. [*α*]^28^_D_ −3.1 (*c* 0.19, MeOH); UV (MeOH) λ_max_ (log ε): 214 (4.1), 228 (4.2), 316 (4.4) nm; IR (film) ν_max_: 3360, 2988, 2902, 1690, 1632, 1605, 1445, 1263, 1171, 1042, 834 cm^–1^; ^1^H NMR (CD_3_OD, 600 MHz) data, see Table 3; ^13^C NMR (CD_3_OD, 100 MHz) data, see Table 4; HRESIMS *m/z* 495.1474 [M + Na]^+^ (calculated for C_21_H_28_NaO_12_, 495.1478).

Compound **4**: white amorphous powder. [*α*]^28^_D_ −26.0 (*c* 0.66, MeOH); UV (MeOH) λ_max_ (log ε): 213 (4.1), 228 (4.2), 317 (4.4) nm; IR (film) ν_max_: 3382, 2925, 1694, 1630, 1604, 1515, 1262, 1169, 1037, 834 cm^–1^; ^1^H NMR (CD_3_OD, 400 MHz) data, see Table 3; ^13^C NMR (CD_3_OD, 100 MHz) data, see Table 4; HRESIMS *m/z* 495.1476 [M + Na]^+^ (calculated for C_21_H_28_NaO_12_, 495.1478).

Compound **5**: white amorphous powder. [*α*]^27^_D_ −13.2 (*c* 0.32, MeOH); UV (MeOH) λ_max_ (log ε): 214 (4.1), 227 (4.2), 316 (4.4) nm; IR (film) ν_max_: 3361, 2922, 1686, 1632, 1604, 1448, 1204, 1171, 1040, 833 cm^–1^; ^1^H NMR (CD_3_OD, 400 MHz) data, see Table 3; ^13^C NMR (CD_3_OD, 100 MHz) data, see Table 4; HRESIMS *m/z* 641.2057 [M + Na]^+^ (calculated for C_27_H_38_NaO_16_, 641.2058).

### 3.4. Acid Hydrolysis of Compounds ***1***–***5***

Compounds **1**–**5** (2 mg), dissolved with 0.1 mL MeOH, were added into 2 mL H_2_SO_4_ aqueous solution (1 M) and kept at 95 °C for 6 h. Then, 2 mL Ba(OH)_2_ solution (1 M) was injected. The hydrolyzed solution was percolated and condensed. The monosaccharides in the concentrated solution were confirmed by TLC (EtOAc-MeOH-HOAc-H_2_O, 8:1:1:0.7, 2 developments) with authentic samples [4]. The *R_f_* values of D-glucose and L-rhamnose were 0.43 and 0.73, respectively.

### 3.5. Determination of Bioactivities 

The inhibitory activities on FAS, α-glucosidase, and α-amylase and the DPPH and ABTS radical scavenging effects of compounds **1**–**10** were tested by previously published methods [4,15,27,28], while orlistat, acarbose, and l-(+)-ascorbic acid were used as positive controls (Appendix A). 

### 3.6. Statistical Analyses 

The statistical analyses were executed using GraphPad Prism 5.01. Every sample was tested in triplicate. The IC_50_ value of a compound (the ultimate concentration of a compound needed to inhibit 50% of the enzyme activity or clear away 50% of the free radicals) was obtained by plotting the inhibition or scavenging percentage of every sample of the compound against its concentration. The results are expressed as the mean ± standard deviation (SD). The difference of the means between groups was analyzed by one-way analysis of variance (ANOVA) using the statistical package SPSS 25.0. The difference between groups was considered to be significant when *p* < 0.05.

## 4. Conclusions

In summary, nine novel compounds, including two hexenol glycosides (**1a** and **1b**), two butenol glycosides (**2a** and **2b**), and five sugar esters (**3a**, **3b**, **4b**, **5a**, and **5b**), together with seven known compounds (**4a** and **6**–**10**), were isolated from the leaves of *L. robustum* and identified with spectroscopic methods (i.e., ^1^H, ^13^C NMR, ^1^H-^1^H COSY, HSQC, HMBC, and HRESIMS) and a chemical method. The biological assays showed that the FAS inhibitory activity of compound **2** (IC_50_ 4.10 ± 0.12 μM) was as strong as the positive control orlistat (IC_50_ 4.46 ± 0.13 μM); the α-glucosidase inhibitory activities of compounds **7** and **9** and the α-amylase inhibitory activities of compounds **1**–**10** were moderate; the DPPH radical scavenging effects of compound **6** (IC_50_ 46.66 ± 1.58 μM) were weaker than l-(+)-ascorbic acid (IC_50_ 13.66 ± 0.13 μM); the ABTS radical scavenging effects of compounds **1** and **10** (IC_50_ 3.41 ± 0.08~5.65 ± 0.19 μM) were more potent than the positive control l-(+)-ascorbic acid (IC_50_ 10.06 ± 0.19 μM), while the ABTS radical scavenging effects of compounds **3**, **4**, **7**, and **9** (IC_50_ 8.78 ± 0.09~12.04 ± 0.08 μM) were as strong as l-(+)-ascorbic acid. Based on this work and previous studies [4,15,16], phenylethanoid, phenylmethanoid, monoterpenoid, hexenol, and butenol glycosides, together with sugar esters, are considered as the main active constituents of *L. robustum* for the prevention of diabetes and its complications. This study provides a theoretical foundation for the leaves of *L. robustum* as a functional tea to prevent diabetes and its complications. It is well known, however, that the effect of a compound in vitro is not necessarily equal to its actual effect in vivo. Therefore, further study should be performed to evaluate the activity of the isolates in vivo in the future.

## Figures and Tables

**Figure 1 molecules-28-00362-f001:**
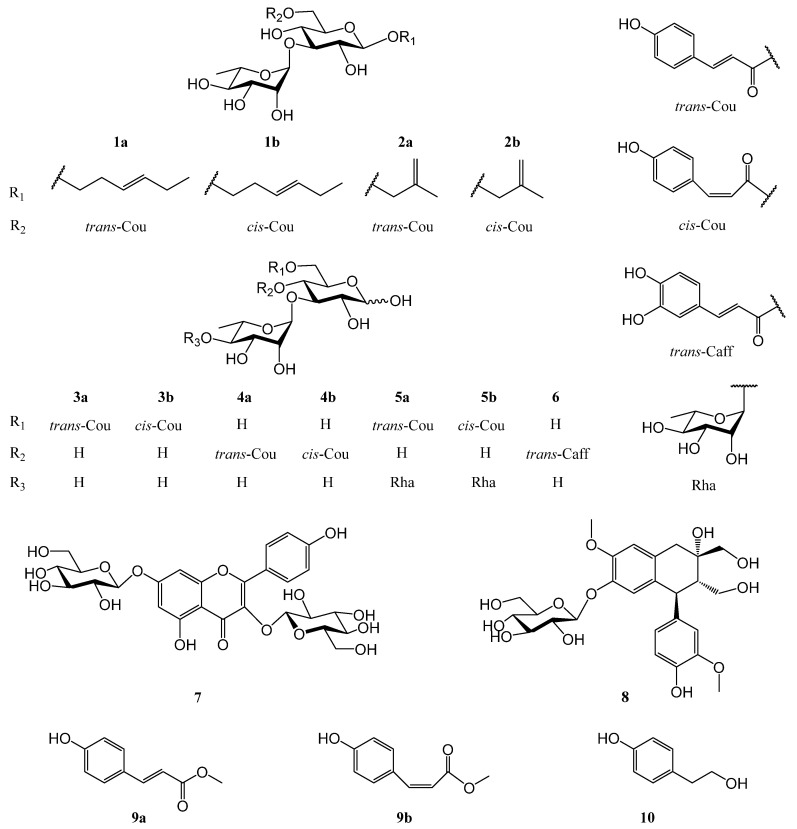
Structures of compounds **1**–**10** from the leaves of *L. robustum*.

**Figure 2 molecules-28-00362-f002:**
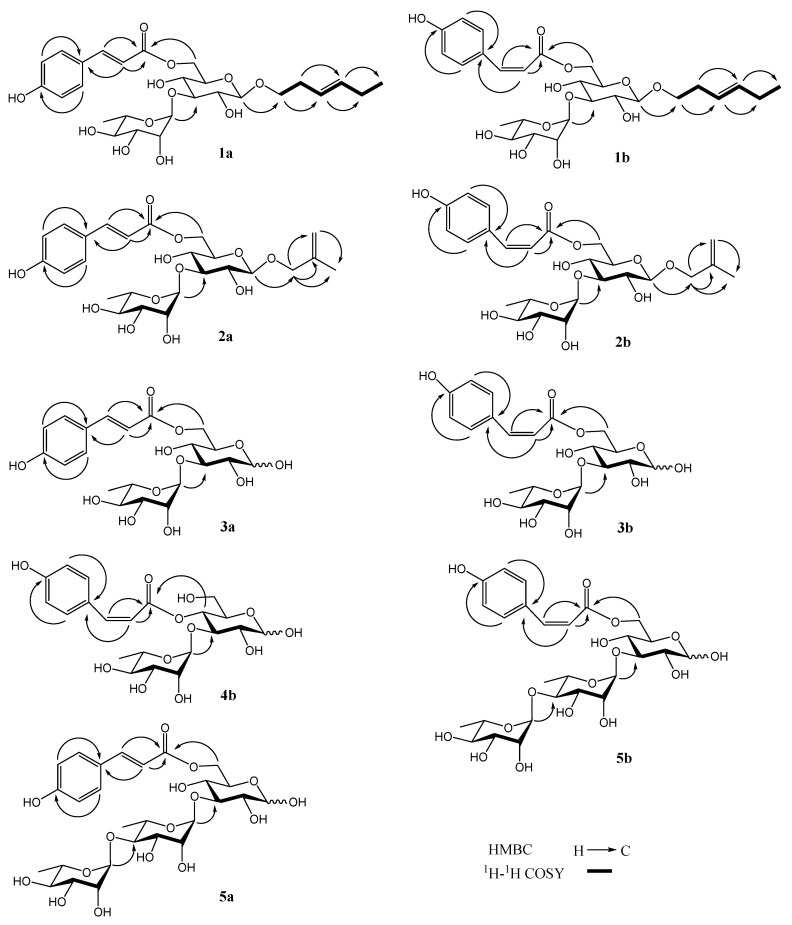
Key HMBC and ^1^H-^1^H COSY correlations of compounds **1**–**5**.

**Table 1 molecules-28-00362-t001:** ^1^H NMR (600 MHz) data of compounds **1**–**2** from *L. robustum* in CD_3_OD *^a^*.

No.	1a	1b	2a	2b
1a	3.55 m	3.55 m	4.07 d (12.6)	4.10 d (12.6)
1b	3.80 m	3.80 m	4.20 d (12.6)	4.15 d (12.6)
2	2.37 m	2.37 m		
3a	5.36 dt (17.4, 6.6)	5.36 dt (17.4, 6.6)	4.88 br. s	4.88 br. s
3b			5.02 br. s	5.02 br. s
4	5.42 dt (17.4, 6.6)	5.42 dt (17.4, 6.6)	1.75 s	1.73 s
5	2.05 m	2.05 m		
6a	0.93 t (7.2)	0.93 t (7.2)		
6b	0.97 t (7.2)	0.97 t (7.2)		
Glc				
1′	4.31 d (8.4)	4.27 d (7.8)	4.30 d (7.2)	4.26 d (7.8)
2′	3.30 m	3.30 m	3.34 m	3.34 m
3′	3.51 m	3.51 m	3.52 m	3.52 m
4′	3.40 t (9.6)	3.40 t (9.6)	3.42 br. d (9.0)	3.42 br. d (9.0)
5′	3.54 m	3.54 m	3.52 m	3.52 m
6′a	4.35 dd (12.0, 6.0)	4.34 dd (12.0, 6.0)	4.36 dd (12.0, 6.0)	4.36 dd (12.0, 6.0)
6′b	4.48 dd (12.0, 2.4)	4.46 dd (12.0, 2.4)	4.48 dd (12.0, 1.8)	4.46 dd (12.0, 1.8)
Rha				
1″	5.18 d (1.8)	5.16 d (1.8)	5.18 d (1.8)	5.16 d (1.8)
2″	3.94 m	3.94 m	3.94 dd (3.6, 1.8)	3.94 dd (3.6, 1.8)
3″	3.71 dd (9.6, 3.6)	3.71 dd (9.6, 3.6)	3.70 dd (9.6, 3.6)	3.70 dd (9.6, 3.6)
4″	3.39 t (9.6)	3.39 t (9.6)	3.40 br. d (9.6)	3.40 br. d (9.6)
5″	4.00 m	4.00 m	4.00 m	4.00 m
6″	1.25 d (6.0)	1.24 d (6.0)	1.25 d( 6.6)	1.25 d( 6.6)
Cou				
2′″	7.43 d (8.4)	7.65 d (8.4)	7.47 d (8.4)	7.65 d (8.4)
3′″	6.77 d (8.4)	6.75 d (8.4)	6.80 d (8.4)	6.76 d (8.4)
5′″	6.77 d (8.4)	6.75 d (8.4)	6.80 d (8.4)	6.76 d (8.4)
6′″	7.43 d (8.4)	7.65 d (8.4)	7.47 d (8.4)	7.65 d (8.4)
7′″	7.63 d (15.6)	6.88 d (13.2)	7.65 d (16.2)	6.89 d (12.6)
8′″	6.33 d (15.6)	5.79 d (13.2)	6.37 d (16.2)	5.80 d (12.6)

*^a^* Coupling constants (*J* values in Hz) are shown in parentheses.

**Table 2 molecules-28-00362-t002:** ^13^C NMR (150 MHz) data of compounds **1**–**2** from *L. robustum* in CD_3_OD.

No.	1a	1b	2a	2b
1	70.8	70.7	74.0	73.8
2	28.9	28.9	143.1	143.1
3	125.8	125.8	113.4	113.4
4	134.6	134.6	19.7	19.7
5	21.5	21.5		
6	14.6	14.6		
Glc				
1′	104.4	104.2	103.0	103.0
2′	75.6	75.6	75.7	75.7
3′	84.0	84.0	84.0	84.0
4′	70.5	70.4	70.4	70.4
5′	75.6	75.3	75.4	75.4
6′	64.6	64.5	64.6	64.6
Rha				
1″	102.7	102.8	102.8	102.8
2″	72.4	72.4	72.4	72.4
3″	72.3	72.3	72.3	72.3
4″	74.0	74.0	74.0	74.0
5″	70.0	70.0	70.0	70.0
6″	17.9	17.9	17.9	17.9
Cou				
1′″	126.3	127.5	126.9	127.5
2′″	131.3	133.8	131.2	133.8
3′″	117.4	116.0	116.9	115.9
4′″	163.0	160.4	161.6	160.2
5′″	117.4	116.0	116.9	115.9
6′″	131.3	133.8	131.2	133.8
7′″	147.1	145.3	146.9	145.3
8′″	114.1	116.2	114.8	116.2
CO	169.2	168.1	169.1	168.1

**Table 3 molecules-28-00362-t003:** ^1^H NMR data of compounds **3**–**5** from *L. robustum* in CD_3_OD *^a^*.

No.	3a *^b^*		3b *^b^*		4b *^c^*
	β	α	β	α	β
Glc					
1	4.52 d (7.8)	5.08 d (3.6)	4.49 d (7.8)	5.06 d (4.2)	4.52 d (7.6)
2	3.27 m	3.49 dd (9.6, 3.6)	3.26 m	3.48 dd (9.6, 4.2)	3.33 m
3	3.53 t (9.6)	3.81 t (9.6)	3.52 t (9.0)	3.77 t (9.6)	3.75 t (9.2)
4	3.40 m	3.41 m	3.39 m	3.40 m	4.85 t (9.2)
5	3.58 m	4.08 dd (9.6, 3.6)	3.57 m	4.07 dd (9.6, 3.6)	3.55 m
6a	4.36 dd (12.0, 6.0)	4.32 dd (12.0, 3.6)	4.26 dd (12.0, 5.4)	4.26 dd (12.0, 3.6)	3.52 m
6b	4.45 dd (12.0, 1.8)	4.49 dd (12.0, 1.8)	4.39 dd (12.0, 1.8)	4.45 dd (12.0, 1.8)	3.58 m
Rha					
1′	5.18 d (1.8)	5.13 d (1.8)	5.15 d (1.8)	5.10 d (1.8)	5.12 d (2.0)
2′	3.97 m	3.97 m	3.96 m	3.96 m	3.93 m
3′	3.72 m	3.72 m	3.71 m	3.71 m	3.58 m
4′	3.41 m	3.41 m	3.40 m	3.40 m	3.32 m
5′	4.02 dd (9.6, 6.0)	4.02 dd (9.6, 6.0)	4.01 dd (9.6, 6.0)	4.01 dd (9.6, 6.0)	3.63 m
6′	1.26 d (6.0)	1.26 d (6.0)	1.25 d (6.0)	1.25 d (6.0)	1.17 d (6.0)
Cou					
2″	7.45 d (8.4)	7.45 d (8.4)	7.66 d (7.8)	7.66 d (7.8)	7.72 d (8.8)
3″	6.80 d (8.4)	6.80 d (8.4)	6.75 d (7.8)	6.75 d (7.8)	6.76 d (8.8)
5″	6.80 d (8.4)	6.80 d (8.4)	6.75 d (7.8)	6.75 d (7.8)	6.76 d (8.8)
6″	7.45 d (8.4)	7.45 d (8.4)	7.66 d (7.8)	7.66 d (7.8)	7.72 d (8.8)
7″	7.63 d (16.2)	7.63 d (16.2)	6.86 d (13.2)	6.86 d (13.2)	6.94 d (12.8)
8″	6.33 d (16.2)	6.33 d (16.2)	5.76 d (13.2)	5.76 d (13.2)	5.81 d (12.8)
**No.**	**4b *^c^***	**5a *^c^***		**5b *^c^***	
	**α**	**β**	**α**	**β**	**α**
Glc					
1	5.11 d (3.6)	4.51 d (8.0)	5.07 d (3.6)	4.51 d (8.0)	5.06 d (3.6)
2	3.56 m	3.26 m	3.48 m	3.26 m	3.48 m
3	4.06 t (9.2)	3.53 m	3.81 t (9.2)	3.53 m	3.81 t (9.2)
4	4.88 t (9.2)	3.40 m	3.40 m	3.40 m	3.40 m
5	4.01 m	3.56 m	4.07 m	3.56 m	4.07 m
6a	3.52 m	4.33 dd (12.0, 5.6)	4.30 dd (12.0, 6.0)	4.33 dd (12.0, 5.6)	4.30 dd (12.0, 6.0)
6b	3.58 m	4.45 dd (12.0, 2.0)	4.50 dd (12.0, 2.0)	4.45 dd (12.0, 2.0)	4.50 dd (12.0, 2.0)
Inner-Rha					
1′	5.17 d (2.0)	5.19 d (1.6)	5.13 d (1.6)	5.17 d (1.6)	5.11 d (1.6)
2′	3.93 m	3.91 m	3.91 m	3.91 m	3.91 m
3′	3.58 m	3.61 dd (9.6, 3.2)	3.85 dd (9.2, 3.2)	3.61 dd (9.6, 3.2)	3.85 dd (9.2, 3.2)
4′	3.32 m	3.54 m	3.54 m	3.54 m	3.54 m
5′	3.63 m	4.12 dd (9.6, 6.0)	4.12 dd (9.6, 6.0)	4.12 dd (9.6, 6.0)	4.12 dd (9.6, 6.0)
6′	1.16 d (6.0)	1.29 d (6.0)	1.29 d (6.0)	1.29 d (6.0)	1.29 d (6.0)
Outer-Rha					
1″		5.20 d (1.6)	5.20 d (1.6)	5.20 d (1.6)	5.20 d (1.6)
2″		3.95 dd (3.2, 1.6)	3.95 dd (3.2, 1.6)	3.95 dd (3.2, 1.6)	3.95 dd (3.2, 1.6)
3″		3.61 dd (9.6, 3.2)	3.61 dd (9.6, 3.2)	3.61 dd (9.6, 3.2)	3.61 dd (9.6, 3.2)
4″		3.40 m	3.40 m	3.40 m	3.40 m
5″		3.72 dd (9.2, 6.0)	3.72 dd (9.2, 6.0)	3.72 dd (9.2, 6.0)	3.72 dd (9.2, 6.0)
6″		1.25 d (6.0)	1.25 d (6.0)	1.25 d (6.0)	1.25 d (6.0)
Cou					
2′″	7.72 d (8.8)	7.46 d (8.4)	7.46 d (8.4)	7.64 d (8.4)	7.63 d (8.4)
3′″	6.76 d (8.8)	6.81 d (8.4)	6.81 d (8.4)	6.76 d (8.4)	6.75 d (8.4)
5′″	6.76 d (8.8)	6.81 d (8.4)	6.81 d (8.4)	6.76 d (8.4)	6.75 d (8.4)
6′″	7.72 d (8.8)	7.46 d (8.4)	7.46 d (8.4)	7.64 d (8.4)	7.63 d (8.4)
7′″	6.95 d (12.8)	7.64 d (16.0)	7.64 d (16.0)	6.87 d (12.8)	6.87 d (12.8)
8′″	5.80 d (12.8)	6.35 d (16.0)	6.34 d (16.0)	5.79 d (12.8)	5.78 d (12.8)

*^a^* Coupling constants (*J* values in Hz) are shown in parentheses. *^b^* At 600 MHz. *^c^* At 400 MHz.

**Table 4 molecules-28-00362-t004:** ^13^C NMR (100 MHz) data of compounds **3**-**5** from *L. robustum* in CD_3_OD.

No.	3a		3b		4b		5a		5b	
	β	α	β	α	β	α	β	α	β	α
Glc										
1	98.1	94.0	98.1	94.1	98.2	94.0	98.1	94.1	98.1	94.1
2	76.8	74.2	76.7	74.2	77.3	74.6	77.0	74.4	77.0	74.4
3	84.1	81.7	84.2	81.8	81.9	79.4	83.6	81.3	83.6	81.3
4	70.6	70.4	70.7	70.5	70.6	70.5	70.6	70.4	70.6	70.4
5	75.4	70.8	75.3	70.8	76.1	71.2	75.5	70.9	75.5	70.9
6	64.8	64.8	64.6	64.6	62.4	62.5	64.9	64.9	64.9	64.9
Inner-Rha										
1′	102.7	102.8	102.9	102.9	103.1	103.2	102.4	102.6	102.4	102.6
2′	72.3	72.3	72.3	72.3	72.3	72.3	72.9	72.9	72.9	72.9
3′	72.2	72.2	72.2	72.2	72.1	72.0	72.9	73.1	72.9	73.1
4′	74.0	74.0	74.1	74.0	73.8	73.8	81.2	81.1	81.2	81.1
5′	70.0	70.0	70.0	70.0	70.4	70.4	68.4	68.4	68.4	68.4
6′	17.9	17.9	17.9	17.9	18.2	18.2	18.6	18.6	18.6	18.6
Outer-Rha										
1″							103.2	103.2	103.2	103.2
2″							72.4	72.4	72.4	72.4
3″							72.4	72.4	72.4	72.4
4″							73.9	73.9	73.9	73.9
5″							70.4	70.4	70.4	70.4
6″							17.8	17.8	17.8	17.8
Cou										
1′″	126.9	126.9	127.5	127.5	127.5	127.5	127.2	127.1	127.5	127.5
2′″	131.1	131.1	133.7	133.7	134.3	134.3	131.2	131.2	133.8	133.8
3′″	116.9	116.9	115.9	115.9	115.8	115.9	116.8	116.8	115.9	115.9
4′″	161.6	161.6	160.2	160.2	160.4	160.5	161.3	161.3	160.4	160.4
5′″	116.9	116.9	115.9	115.9	115.8	115.9	116.8	116.8	115.9	115.9
6′″	131.1	131.1	133.7	133.7	134.3	134.3	131.2	131.2	133.8	133.8
7′″	146.8	146.8	145.3	145.3	147.1	147.3	146.7	146.8	145.2	145.2
8′″	114.7	114.7	116.2	116.2	116.1	116.1	115.0	114.9	116.3	116.3
CO	169.2	169.1	168.2	168.1	167.0	166.9	169.2	169.1	168.2	168.2

**Table 5 molecules-28-00362-t005:** Results of the bioactivity assays of compounds **1**–**10** from *L. robustum^a^*.

Compound	FAS IC_50_ (μM) *^b^*	α-Glucosidase Inhibition at 0.1 mM (% )	α-Amylase Inhibition at 0.1 mM (%)	DPPH IC_50_ (μM) *^b^*	ABTS^•+^ IC_50_ (μM) *^b^*
**1**	NA *^c^*	NA	27.9 ± 6.4 bc	NA	5.65 ± 0.19 b
**2**	4.10 ± 0.12 a	NA	24.0 ± 1.5 bc	NA	103.4 ± 4.00 g
**3**	6.25 ± 0.20 b	NA	29.8 ± 1.8 bc	>250	12.04 ± 0.08 d
**4**	10.49 ± 0.32 e	NA	25.6 ± 1.0 bc	NA	11.21 ± 0.40 cd
**5**	9.75 ± 0.24 d	NA	26.5 ± 4.0 bc	>250	15.54 ± 0.36 e
**6**	NA	NA	23.0 ± 0.7 c	46.66 ± 1.58 b	17.01 ± 0.45 e
**7**	8.10 ± 0.37 c	15.6 ± 0.9 c	31.8 ± 0.5 b	NA	9.34 ± 0.04 cd
**8**	8.01 ± 0.26 c	NA	28.5 ± 2.7 bc	>250	29.13 ± 1.11 f
**9**	15.41 ± 0.42 f	33.8 ± 2.9 b	29.5 ± 0.6 bc	>250	8.78 ± 0.09 c
**10**	NA	NA	16.2 ± 5.0 d	NA	3.41 ± 0.08 a
Orlistat *^d^*	4.46 ± 0.13 a				
Acarbose *^d^*		93.2 ± 0.1 a	51.8 ± 2.5 a		
l-(+)-ascorbic acid *^d^*				13.66 ± 0.13 a	10.06 ± 0.19 cd

*^a^* Data are expressed as the mean ± SD (*n* = 3). Means with the same letter are not significantly different (one-way analysis of variance, α = 0.05). *^b^* IC_50_: the ultimate concentration of sample needed to inhibit 50% of the enzyme activity or clear away 50% of the free radicals. *^c^*NA: no activity. *^d^*Positive control.

## Data Availability

The data presented in this study are available in the Appendix A.

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
