# Peer review of "Chemical Constituents from the Leaves of Ligustrum robustum and Their Bioactivities"

_molecules, 2023, doi:10.3390/molecules28010362_

Round 1

Reviewer 1 Report

The authors reported nine undescribed glycosides and seven known compounds from the leaves of Ligustrum robustum, including two new hexenol glycosides, two new butenol glycosides, and five new sugar esters. Their inhibitory effects on a-glucosidase, a-amylase, fatty acid synthase (FAS) and the antioxidant effects were evaluated. Some compounds showed inhibitory activity. The manuscript could be published after major revision.

1. In page 3, line 68, the methyl signal at 0.93 should be assigned to 1a, while the methyl signal at 0.97 should be assigned to 1b. The same problem occurred in Table 2.

2. The acid hydrolysis experiment of 1 gave D-glucose and L-rhamnose affirmed by TLC. But, the absolute configuration of monosacchride cant be determined by TLC, because D-glucose and L-glucose have the same chromatography behavior on TLC.

3. In page 6, lines 130-131, 5.79, 6.88 (1H each, d, J = 13.2 Hz, H-7''', H-8''')” should be 5.79, 6.88 (1H each, d, J = 13.2 Hz, H-8''', H-7''')”. The similar problem was in 2b (line 170).

4. Compounds 1, 2, 3 and 4 were obtained as mixture with trans- p-coumaroyl and cis-p-coumaroyl isomers. Could they be isolated?

Author Response

  1. In page 3, line 68, the methyl signal at 0.93 should be assigned to 1a, while the methyl signal at 0.97 should be assigned to 1b. The same problem occurred in Table 2.

Authors accept the advice and revise as follows: and two methyl groups at dH 0.93 (2H, t, J = 7.2 Hz, 6a), 0.97 (1H, t, J = 7.2Hz, 6b) and 1.25 (3H, d, J = 6.0 Hz). 

  1. The acid hydrolysis experiment of 1gave D-glucose and L-rhamnose affirmed by TLC. But, the absolute configuration of monosacchride can’t be determined by TLC, because D-glucose and L-glucose have the same chromatography behavior on TLC.

Authors accept the advice and revise as follows: The acid hydrolysis experiment of 1 gave D-glucose and L-rhamnose affirmed by TLC and comparison of its NMR data with those of ligurobustoside E [12].

  1. In page 6, lines 130-131, “5.79, 6.88 (1H each, d, J = 13.2 Hz, H-7''', H-8''')” should be “5.79, 6.88 (1H each, d, J = 13.2 Hz, H-8''', H-7''')”. The similar problem was in 2b (line 170).

Authors accept the advice.

  1. Compounds 1, 2, 3and 4were obtained as mixture with trans- p-coumaroyl and cis-p-coumaroyl isomers. Could they be isolated?

They could be isolated by AgNO3 silica gel column chromatography when the weight of the mixture was more than 20 mg.

Reviewer 2 Report

The article entitled "Chemical constituents from the leaves of Ligustrum robustum and their bioactivities" by Lu et al. has allowed the isolation and identification of 9 novel bioactive compounds from Ligustrum robustum. The article can be interesting; however, it has some major weaknesses that should be addressed before its publication in Molecules. In general, the article should be completed with a broader bibliographic search that allows a better contextualization of the field of study, as well as discussing and comparing the results obtained in any way. Moreover, it should also be noted that the references used for its preparation are scarce and less than 30% of these are references from the last 5 years.

Abstract

Line 15-24: The "Abstract" section does not provide a summary of the article. A background is not included, nor are the methods carried out for its conduction specified. As indicated in the guide for authors of Molecules, the "Abstract" should have a 1) Background; 2) Methods; 3) Results; and 4) Conclusions, (not including headings). Rephrase the abstract accordingly.

Introduction

The introduction is too short since it hardly places the study in the context, and it does not delve into the current state of the field of research. The references are scarce, especially considering that 10 of the 17 references used in the Introduction section refer to only one statement. I think that this part of the text could be extended and completed by relying on other extra references and/ or detailing the bibliography already cited to a greater extent.

Line 38: According to the authors' guide, the meaning of FAS must be specified since it is the first time it is cited in the main text. Note that Acronyms/Abbreviations/Initialisms should be defined the first time they appear in each of three sections: the abstract; the main text; the first figure or table. Revise the rest of the manuscript accordingly.

Results and Discussion

Line 54-59: Why add this sentence? It makes no sense in the "Results and Discussion" part since it is describing the methodology used in the extraction. Delete that paragraph.

Lines 283-295: It is too long a sentence. There is no point in 13 lines. Consider rewriting this sentence by adding a period.

Lines 296-300, Table 5: Consider adding the letters that indicate significant differences as superscript/ subscript, to be more visual. Also consider adding an additional column at the end of the Table 5 indicating the P-value or significance levels (for instance, *** P < 0.001; ** P < 0.01; * P < 0.05; ns: no significance), indicating these in the table foot.

Lines 301-306: Actually, no discussion of the bioactivity of Ligustrum robustum compounds takes place. Could some functional group (such as -OH groups) and delocalized electrons be related to some of the bioactive properties of the detected compounds (such as antioxidant activity)? This could be a good moment to discuss these aspects and to relate the chemical composition and the structure of the new compounds identified with their bioactivity.

Materials and Methods

Lines 394-407: Consider briefly describing the methods for determining bioactivities (i.e., FAS, α-glucosidase and α-amylase, and the DPPH and ABTS radical scavenging) and not simply providing the references.

Author Response

  1. It should also be noted that the references used for its preparation are scarce and less than 30% of these are references from the last 5 years.

     Authors accept the advice and add 5 references published in 2022.

  1. Line 15-24: The "Abstract" section does not provide a summary of the article. A background is not included, nor are the methods carried out for its conduction specified. As indicated in the guide for authors of Molecules, the "Abstract" should have a 1) Background; 2) Methods; 3) Results; and 4) Conclusions, (not including headings). Rephrase the abstract accordingly.

 Authors accept the advice and revise as follows: The leaves of Ligustrum robustum have been consumed as Ku-Ding-Cha for clearing heat and removing toxins, and used as a folk medicine for curing hypertension, diabetes and obesity in China. The phytochemical research on the leaves of L. robustum led to the isolation and identification of two new hexenol glycosides, two new butenol glycosides, and five new sugar esters, named ligurobustosides X (1a), X1 (1b), Y (2a), Y1 (2b) and ligurobustates A (3a), B (3b), C (4b), D (5a), E (5b), along with seven known compounds (4a, 6-10). Compounds 1-10 were tested for the inhibitory effects on fatty acid synthase (FAS), a-glucosidase, a-amylase, and the antioxidant activities. Compound 2 showed strong FAS inhibitory activity (IC50: 4.10 ± 0.12 μM) as the positive control orlistat (IC50: 4.46 ± 0.13 μM); compounds 7 and 9 revealed moderate a-glucosidase inhibitory activities; compounds 1-10 showed moderate a-amylase inhibitory activities; compounds 1 and 10 displayed stronger 2,2'-azino-bis(3- ethylbenzthiazoline-6-sulphonic acid) ammonium salt (ABTS) radical scavenging effects (IC50: 3.41 ± 0.08~5.65 ± 0.19 μM) than the positive control L-(+)-ascorbic acid (IC50: 10.06 ± 0.19 μM). This study provided a theoretical foundation for the leaves of L. robustum as a functional tea to prevent obesity and diabetes. 

  1. The introduction is too short since it hardly places the study in the context, and it does not delve into the current state of the field of research. The references are scarce, especially considering that 10 of the 17 references used in the Introduction section refer to only one statement. I think that this part of the text could be extended and completed by relying on other extra references and/ or detailing the bibliography already cited to a greater extent.

Authors accept the advice and revise as follows:

Diabetes, which affected nearly 10.5% of the population in the world, is a chronic metabolic disease characterized by hyperglycemia caused by insulin resistance, deficiency in insulin secretion, or both [1]. Its complications, including diabetic neuropathy, nephropathy and cardiovascular diseases, lead to serious morbidity and mortality [1]. The present drugs, such as insulin, metformin, sulfonylureas, acarbose, could control hyperglycemia, but the effect of preventing the complications of diabetes was not ideal. Therefore, it is significant to search new resource of preventing diabetes and its complications. 

Studies revealed that, long-term obesity might trigger specific metabolic disorders, like cardiovascular diseases, insulin resistance and diabetes [2,3]; fatty acid synthase (FAS), which catalyzed the synthesis of saturated long-chain fatty acids, was a potential target to prevent obesity [4]; carbohydrate digestive enzymes, such as a-glucosidase and a-amylase, played a crucial role in promoting hyperglycemia by releasing monosaccharides in the course of digestion [5]; the contribution of reactive oxygen species generated by the oxidative stress induced by chronic hyperglycemia was linked to the onset and progression of diabetes and its complications [6]. Thus, the natural products with inhibitory activities on FAS, a-glucosidase, a-amylase, and antioxidant effect might be a new resource to prevent diabetes and its complications.

  1. Line 38: According to the authors' guide, the meaning of FAS must be specified since it is the first time it is cited in the main text. Note that Acronyms/Abbreviations/Initialisms should be defined the first time they appear in each of three sections: the abstract; the main text; the first figure or table. Revise the rest of the manuscript accordingly.

Authors accept the advice and revise as follows: Studies revealed that, long-term obesity might trigger specific metabolic disorders, like cardiovascular diseases, insulin resistance and diabetes [2,3]; fatty acid synthase (FAS), which catalyzed the synthesis of saturated long-chain fatty acids, was a potential target to prevent obesity [4]. 

  1. Line 54-59: Why add this sentence? It makes no sense in the "Results and Discussion" part since it is describing the methodology used in the extraction. Delete that paragraph.

Authors accept the advice.

  1. Lines 283-295: It is too long a sentence. There is no point in 13 lines. Consider rewriting this sentence by adding a period.

Authors accept the advice.

  1. Lines 296-300, Table 5: Consider adding the letters that indicate significant differences as superscript/ subscript, to be more visual. Also consider adding an additional column at the end of the Table 5 indicating the P-value or significance levels (for instance, *** P< 0.001; ** P < 0.01; * P < 0.05; ns: no significance), indicating these in the table foot.

Authors don’t accept the advice. Table 5 was designed properly for one-way analysis of variance (but not for t-test). We referred to the published tables (Lu, S.-H. Monoterpenoid glycosides from the leaves of Ligustrum robustum and their bioactivities. Molecules 202227, 3709; Lu, S.-H. Phenylethanoid and phenylmethanoid glycosides from the leaves of Ligustrum robustum and their bioactivities. Molecules 2022, 27, 7390). 

  1. Lines 301-306: Actually, no discussion of the bioactivity of Ligustrum robustum compounds takes place. Could some functional group (such as -OH groups) and delocalized electrons be related to some of the bioactive properties of the detected compounds (such as antioxidant activity)? This could be a good moment to discuss these aspects and to relate the chemical composition and the structure of the new compounds identified with their bioactivity.

Authors accept the advice and revise as follows: From the results of DPPH and ABTS assays, the phenolic hydroxy group in a compound was believed as a key factor for the antioxidant effect.

  1. Lines 394-407: Consider briefly describing the methods for determining bioactivities (i.e., FAS, a-glucosidase and a-amylase, and the DPPH and ABTS radical scavenging) and not simply providing the references.

The methods were offered as Supplementary Materials S1.

Round 2

Reviewer 2 Report

My suggestion has been correctly addressed.